# Improving Visual Prompt Tuning by Gaussian Neighborhood Minimization for Long-Tailed Visual Recognition

**Mengke Li**
Guangming Laboratory
Shenzhen, China
limengke@gml.ac.cn

**Ye Liu**
Guangming Laboratory
Shenzhen, China
zbdly226@gmail.com

**Yang Lu**
Xiamen University
Xiamen, China
luyang@xmu.edu.cn

**Yiqun Zhang**
Guangdong University of Technology
Guangzhou, China
yqzhang@gdut.edu.cn

**Yiu-ming Cheung**
Hong Kong Baptist University
Hong Kong SAR, China
ymc@comp.hkbu.ed

**Hui Huang***
Shenzhen University
Shenzhen, China
hhzhiyan@gmail.com

## Abstract

Long-tailed visual recognition has received increasing attention recently. Despite fine-tuning techniques represented by visual prompt tuning (VPT) achieving substantial performance improvement by leveraging pre-trained knowledge, models still exhibit unsatisfactory generalization performance on tail classes. To address this issue, we propose a novel optimization strategy called Gaussian neighborhood minimization prompt tuning (GNM-PT), for VPT to address the long-tail learning problem. We introduce a novel Gaussian neighborhood loss, which provides a tight upper bound on the loss function of data distribution, facilitating a flattened loss landscape correlated to improved model generalization. Specifically, GNM-PT seeks the gradient descent direction within a random parameter neighborhood, independent of input samples, during each gradient update. Ultimately, GNM-PT enhances generalization across all classes while simultaneously reducing computational overhead. The proposed GNM-PT achieves state-of-the-art classification accuracies of 90.3%, 76.5%, and 50.1% on benchmark datasets CIFAR100-LT (IR 100), iNaturalist 2018, and Places-LT, respectively. The source code is available at https://github.com/Keke921/GNM-PT.

## 1 Introduction

Long-tailed visual recognition provides solutions to the challenges posed by the prevalent imbalance and multitude of classes in real-world data. Its training data mirror the real-world distribution, wherein a few categories (head classes) boast abundant samples, while a substantial number of categories (tail classes) exhibit very few samples, conforming to a long-tail distribution [58]. Given its ubiquity and practicality, long-tailed visual recognition has attracted considerable attention, and numerous approaches have been proposed in recent years. Based on the data processing workflow, these methods can be broadly categorized into three types [30]: 1) data manipulation [17, 49, 8, 59, 35], 2) representation improvement [68, 23, 72, 22], and 3) model output modification [2, 48, 42, 32, 33]. These methods address the challenge of long-tailed learning from diverse perspectives by extending the traditional training-from-scratch approach.

---

*Corresponding author.

38th Conference on Neural Information Processing Systems (NeurIPS 2024).

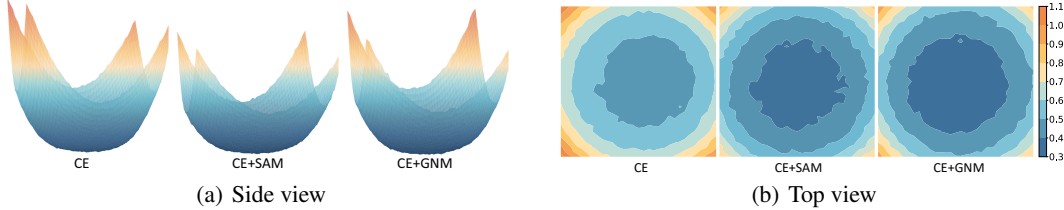

(a) Side view        (b) Top view

Figure 1: Loss landscape comparison of VPT based on ViT-B/16 with CE loss (best view in color). The dataset used is CIFAR100-LT with an imbalance ratio of 100.

Recently, leveraging the robust discriminative capabilities of pre-trained models through the integration of multi-head self-attention (MHSA) based networks [54, 11] and parameter-efficient fine-tuning (PEFT) techniques [21, 69, 63, 15, 16] has led to substantial enhancements in model performance on long-tailed data. For example, Tian et al. [52] introduced the text modality by CLIP [46] to aid in visual representation. Dong et al. [10] exploited visual prompt tuning (VPT) to learn class-shared and group-specific prompts for long-tailed data. These methods essentially increase model compatibility. However, even with the assistance of large-scale pre-trained knowledge, PEFT represented by VPT still exhibits inferior generalization performance on tail classes compared to head classes.

Chen et al. [5] emphasize that converged ViTs exhibit extremely sharp local minima, hindering their generalization [12], particularly for tail classes with limited samples. The imperative necessity to improve tail-class accuracy resides in advancing the generalization capability of PEFT, a facet extensively elucidated within the optimization framework [18]. Searching for flat minima by sharpness-aware minimization (SAM) [14] represents a promising optimization technique to improve model performance (as shown in Figure 1). SAM first captures the sharpness of loss landscape, which correlates with the generalization gap, based on gradient directions, and then searches for flat minima. Nevertheless, SAM encounters two challenges when applied to long-tail data: 1) flat minima primarily target head classes [71, 70], and 2) it involves two sequential gradients computation.

This paper proposes a novel optimization strategy, named Gaussian neighborhood minimization prompt tuning (GNM-PT), inspired by SAM of flattening the loss landscape to enhance model generalization. Since the widespread usage, the flexibility of prompt and the suitability amount of trainable parameters for visualizing the loss landscape, we select VPT as a representative of PEFT technology to study. GNM-PT shows superior performance than SAM-based methods, particularly targeting long-tailed visual recognition tasks.

Specifically, we propose to minimize a novel Gaussian neighborhood loss named Gaussian neighborhood minimization (GNM) to obtain flat minima, substantiated by rigorous theoretical proof. GNM minimizes the mean value of the loss function within the parameter neighborhood, in contrast to the approaches of minimizing the maximum value of the parameter neighborhood employed by SAM [14, 26, 36, 43]. The mean value is a tighter upper bound than the maximum. It is evident from Figure 1(b) that GNM yields a distinctly pronounced convexity, characterized by relatively lower loss values, thereby leading to a more optimal solution. The calculation is achieved by random sampling from a normal distribution as the perturbation for the training parameters. The proposed GNM equally constrains smoothness optimization through a sample-independent perturbation without extra gradient calculations, which eventually improves model performance for long-tailed data. As shown in Figure 1(a), GNM, resembling SAM, flattens the landscape of cross-entropy (CE) loss. To further enhance the classification capability of the PEFT methods exemplified by VPT, we harness information from high-level prompts by merging the prompt with the class token for the ultimate classification. We theoretically validate the rationale behind the proposed method. Extensive experiments on benchmark datasets demonstrate that GNM-PT shows great generalization ability on long-tail data, surpassing existing methods. Ablation experiments further prove that GNM improves model performance as well as maintains computational efficiency. Our main contributions are summarized as follows: 1) We identify pressing concerns, explicitly focusing on the imperative need for pre-trained models to enhance the generalization abilities across all classes while concurrently mitigating computational time. 2) We propose the efficient GNM-PT, tailored for long-tailed visual recognition based on VPT, which can improve model generalization while minimizing computational overhead. 3) We provide theoretical evidence supporting the superiority of the proposed GNM-PT. Comprehensive experiments also demonstrate that GNM-PT outperforms its state-of-the-art counterparts.

## 2  Related work

### 2.1  DNN-Based Model for Long-Tailed Learning

Deep neural networks (DNNs) have made significant advancements in long-tail visual recognition in the last few decades. Re-balancing the data distribution, including re-sampling the input data [3, 40, 23, 56, 64] and re-weighting the loss function [13, 49, 8, 24, 44] is the most direct and effective way to improve the performance of the tail classes. Re-margining methods [2, 33, 42, 32, 55] leave larger margins for tail classes than for head classes to improve the separability of tail classes, which can alleviate overfitting and improve model generalization. However, these methods, while improving the performance of tail classes, come at the cost of sacrificing the accuracy in head classes. Ensembling learning encompasses redundant ensembling [57, 1, 27, 29, 22], which aggregates separate classifiers or networks in a multi-expert framework, and complementary ensembling [68, 6, 61], which involves statistically selecting different data divisions. Studies have demonstrated that ensembling methods, particularly redundant ensembling, can achieve SOTA performance and generate more robust predictions by reducing model variance [27, 57] and/or increasing data diversity [29, 34, 61]. Additionally, various alternative methodologies within the realm of DNN-based long-tailed learning. For example, data augmentation [45, 65], decoupling representation [23, 66], logit adjustment [2, 42, 32, 31], to name a few.

### 2.2  MHSA-Based Fine-Tuning for Long-Tail Learning

Recent advancements in the field of computer vision have harnessed the potential of pre-trained MHSA-Based models, as exemplified by CLIP [46] and the Visual Transformers (ViTs) [21]. In contrast to the conventional practice of training DNNs from scratch, recent proposed PEFT techniques [21, 4, 20], adopted in RAC [38], VL-LTR [52], LPT [10], and PEL [50], to name a few, showcase that the meticulous fine-tuning of pre-trained models can yield surprising improvements in the performance of long-tailed visual recognition tasks. For example, VL-LTR employs a contrastive language-image pre-training approach, known as CLIP, and integrates supplementary image-text web data for fine-tuning. LPT fine-tunes a vision Transformer using visual prompt tuning [21], employing a two-stage training strategy. Nevertheless, it is worth noting that their performance in tail classes still exhibits inferior results compared to that in head classes.

### 2.3  Sharpness-Aware Minimization

Hochreiter and Schmidhuber [18] first pointed out that flat minima corresponds to low network complexity and high generalization performance. Li et al. [28] proposed to visualize the loss landscape and used it to find the flat minima. Subsequently, Foret et al. [14] proposed SAM to seek flat minima and minimize the loss function, so as to improve model generalization capability. Chen et al. [5] demonstrated that ViTs converge at extremely sharp local minima, and they can surpass ResNets in both accuracy and robustness when combined with SAM optimizer. As for long-tailed data, models often showcase varying levels of generalization performance across different classes, with the tail class typically exhibiting inferior performance. Based on this, CCSAM [71] scales the intensity of SAM for classifier inversely with the number of samples available for each class. Zhou et al. [70] observed that, despite the utilization of SAM, the tail classes, due to their substantially smaller sample sizes in comparison to head classes, have limited influence on model parameters. As a result, the loss landscape for tail classes lacks the desired flatness. To address this issue, they proposed ImbSAM, which isolates SAM for head classes and concentrates exclusively on tail classes. Both CCSAM and ImbSAM are designed to bolster generalization capabilities, particularly for tail classes, albeit at the cost of a slight reduction in the performance of head classes.

## 3  Methodology

### 3.1  Preliminaries

**Visual Prompt Tuning.** In VPT [21], $n_p$ prompt tokens $\mathbf{P} = [\mathbf{p}_1, \mathbf{p}_2, \cdots, \mathbf{p}_{n_p}] \in \mathbb{R}^{n_p \times D}$ are trained to facilitate transfer learning on new datasets with a constraint on the number of learnable parameters, where $D$ is the dimension of tokens in the pre-trained ViT [11]. The prompt tokens encode task-specific information through collaboration with patch representations obtained from ViT

blocks. The only parameters that need training are prompt and classification header. There are two variations: 1) VPT-Shallow, which inserts prompts only at the first block, and 2) VPT-Deep, which inserts prompts at all blocks. Take VPT-deep as an example, it is expressed as:

$$\left[\mathbf{z}_{cls}^l, \mathbf{Z}^l\right] = \text{Block}^l\left(\left[\mathbf{p}^{l-1}, \mathbf{z}_{cls}^{l-1}, \mathbf{Z}^{l-1}\right]\right), \tag{1}$$

where $\mathbf{p}^l$ is the learnable prompt, $\mathbf{z}_{cls}^l$ is the class token, and $\mathbf{Z}^l = \left[\mathbf{z}_1^l, \mathbf{z}_2^l, \cdots, \mathbf{z}_{N_z}^l\right]$ represents $N_z$ patch tokens. $\text{Block}^l$ denotes the $l$-th layer of the pre-trained ViT model.

VPT demonstrates notable efficacy in low-data scenarios and maintains its advantages across varying data scales [21]. Despite achieving significant performance improvements, it is noteworthy that VPT exhibits substantial differences in accuracy across different categories. For example, on CIFAR100-LT, the original VPT achieves Top-1 accuracies of 92.11%, 82.86%, and 64.83% for the head, median, and tail classes, respectively. Its generalization ability towards tail classes can be further improved.

**Sharpness-Aware Minimization.** SAM [14] can improve the generalization ability for models by finding an optimal with low curvature. That is, SAM minimizes a specific point and its neighborhoods in the loss landscape of criterion $L_{\mathcal{D}}(\boldsymbol{\theta})$ w.r.t data distribution $\mathcal{D}$. It is derived from PAC-Bayesian generalization bound [41] and following [12, 71], which is, for any $\rho > 0$ and distribution $\mathcal{D}$, with probability $1 - \rho$ over a training set $\mathcal{T}$ i.i.d sampled from $\mathcal{D}$, the criterion $L_{\mathcal{D}}(\boldsymbol{\theta})$ satisfies:

$$L_{\mathcal{D}}(\boldsymbol{\theta}) \leq \max_{\|\boldsymbol{\varepsilon}\|_2^2 \leq \rho} L_{\mathcal{T}}(\boldsymbol{\theta} + \boldsymbol{\varepsilon}) + h(\frac{\|\boldsymbol{\theta}\|_2^2}{\rho^2}), \tag{2}$$

where $h$ is a strictly increasing function. It can be theoretically substituted by an $L_2$ weight decay regularizer $\frac{\lambda}{2}\|\boldsymbol{\theta}\|_2^2$ due to its monotonicity. $\lambda$ denotes the weight decay coefficient. Foret et al. [14] define the sharpness aware loss $L_{\mathcal{T}}^{SAM}(\boldsymbol{\theta}) = \max_{\|\boldsymbol{\varepsilon}\|_2 \leq \rho} L_{\mathcal{T}}(\boldsymbol{\theta} + \boldsymbol{\varepsilon})$ and sharpness of the loss function $L$ as $L_{\mathcal{T}}^{SAM}(\boldsymbol{\theta}) - L_{\mathcal{T}}(\boldsymbol{\theta})$, which measures the loss increasing rate by perturbing $\boldsymbol{\theta}$ with a nearby parameter value $\rho$. They propose a methodology wherein parameter values are selected by solving the sharpness aware minimization (SAM) problem:

$$\boldsymbol{\theta}^* = \min_{\boldsymbol{\theta}} \max_{\|\boldsymbol{\varepsilon}\|_2 \leq \rho} L_{\mathcal{T}}(\boldsymbol{\theta} + \boldsymbol{\varepsilon}) + \frac{\lambda}{2}\|\boldsymbol{\theta}\|_2^2. \tag{3}$$

When comparing Equation (3) to the standard training loss, it requires that the maximum loss value of the parameter within the neighborhood of radius $\rho$ centered on $\boldsymbol{\theta}$ also remains low. The direction of the gradient of $L_{\mathcal{T}}(\boldsymbol{\theta})$ indicates the maximum value of the loss within the neighborhood. Subsequently, for step $t$, the optimal perturbation vector $\hat{\boldsymbol{\varepsilon}}_t$ obtained based on the gradient of $L_{\mathcal{T}(\boldsymbol{\theta})}$ to obtain $L_{\mathcal{T}}^{SAM}$. The parameters are updated w.r.t. the perturbed model parameters $\boldsymbol{\theta} + \hat{\boldsymbol{\varepsilon}}$:

$$\hat{\boldsymbol{\varepsilon}}_t = \rho_{SAM} \frac{\nabla_{\boldsymbol{\theta}} L_{\mathcal{T}}(\boldsymbol{\theta}_t)}{\|\nabla_{\boldsymbol{\theta}} L_{\mathcal{T}}(\boldsymbol{\theta}_t)\|_2^2}, \tag{4}$$

$$\boldsymbol{\theta}_{t+1}^{SAM} = \boldsymbol{\theta}_t - \alpha_t \left(\nabla_{\boldsymbol{\theta}_t} L_{\mathcal{T}}(\boldsymbol{\theta}_t)|_{\boldsymbol{\theta}_t + \hat{\boldsymbol{\varepsilon}}_t} + \lambda \boldsymbol{\theta}_t\right). \tag{5}$$

where $\rho_{SAM}$ represents the radius of the parameter neighborhood for SAM, and $\alpha_t$ denotes the learning rate scheduled in step $t$.

### 3.2 Prompt Tuning with Gaussian Neighborhood Minimization

Despite the effectiveness of SAM and its strong theoretical foundation, it exhibits twofold deficiencies:
● For long-tailed data, $\hat{\boldsymbol{\varepsilon}}$ for tail classes is often negligible due to the dominance of head classes with a large number of samples in determining the gradient direction. Consequently, this leads to a challenge in achieving effective generalization for tail classes [71, 70].
● At each step, two gradient computations are required, namely $\nabla_{\boldsymbol{\theta}} L_{\mathcal{T}}(\boldsymbol{\theta}_t)$ and $\nabla_{\boldsymbol{\theta}} L_{\mathcal{T}}(\boldsymbol{\theta}_t + \hat{\boldsymbol{\varepsilon}}_t)$, resulting in a duplication of the computational overhead.

To address the aforementioned issues of head-class dominant optimization and double gradient computation, we propose GNM-PT.

**Gaussian Neighborhood Minimization (GNM).** To mitigate the presence of sharp minima and enhance the performance of VPT on long-tailed data, we can directly minimize the loss within the

parameter neighborhood, thereby attaining a flattened loss landscape. We introduce the Gaussian neighborhood loss $L_{\mathcal{T}}^{GN}$ on $\mathcal{T}$, which is defined as:

$$L_{\mathcal{T}}^{GN}(\boldsymbol{\theta}) = \mathbb{E}_{\varepsilon_i \in \mathcal{N}(0,\sigma^2)}\left[L_{\mathcal{T}}(\boldsymbol{\theta} + \boldsymbol{\varepsilon})\right]. \tag{6}$$

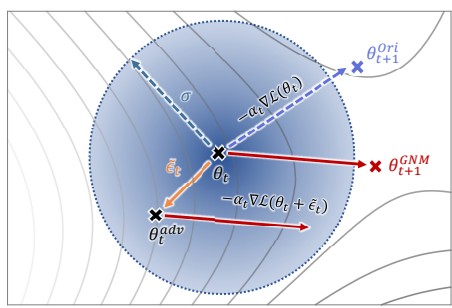

Figure 2: Schematic of optimization direction in GNM[3]. $\boldsymbol{\theta}_{t+1}^{Ori}$ and $\boldsymbol{\theta}_{t+1}^{GNM}$ represent the gradient update w.o. and w. GNM for step $t + 1$.

Optimizing $L_{\mathcal{T}}^{GN}$ is equivalent to optimizing an upper bound of the distribution $\mathcal{D}$ using the training set $\mathcal{T}$ sampled i.i.d. from $\mathcal{D}$. The detailed theoretical proof will be discussed in the following section. Then, substituting $L_{\mathcal{T}}^{SAM}$ in Equation (3) with $L_{\mathcal{T}}^{GN}$, we can obtain the parameter update strategy of GNM:

$$\tilde{\boldsymbol{\varepsilon}}_t = \rho_{GNM} \cdot [\varepsilon_i]_{i=1}^k , \ \varepsilon_i \sim \mathcal{N}(0, \sigma^2), \tag{7}$$

$$\boldsymbol{\theta}_{t+1}^{GNM} = \boldsymbol{\theta}_t - \alpha_t \left(\nabla_{\boldsymbol{\theta}_t} L_{\mathcal{T}}(\boldsymbol{\theta}_t)|_{\boldsymbol{\theta}_t + \tilde{\boldsymbol{\varepsilon}}_t} + \lambda \boldsymbol{\theta}_t\right). \tag{8}$$

$\rho_{GNM}$ in Equation (7) represents the radius of the parameter neighborhood for GNM. The detailed derivation of the gradient for GNM can be found in Appendix A. Figure 2 schematically illustrates a single GNM parameter update.

**Remark 1.** *Handling Long-Tailed Data.* GNM is better suited for long-tailed data.

*Proof.* If $\mathcal{T}$ is a long-tailed training set i.i.d. sampled from $\mathcal{D}$, the direction of gradients in existing methods such as SGD and SAM is predominantly influenced by head classes. (The detailed proof can be found in Appendix B.) Consequently, optimization through Equation (4), which is sample-dependent, will be determined mainly by the head classes. Conversely, Equation (7) is in a sample-independent manner, avoiding classes with large numbers of samples that dominate the direction of the perturbation vector. □

**Remark 2.** *Computational Time.* GNM saves computational overhead compared to SAM.

*Proof.* Even when disregarding second- and higher-order terms (for example, see Foret et al. [14], Zhou et al. [71], Mi et al. [43] for more details), it is apparent from Equation (4) that solving for $\hat{\varepsilon}_t$ necessitates the computation of one gradient involving a forward and backward pass, while calculating $\boldsymbol{\theta}_{t+1}$ requires another forward and backward pass. As a result, in SAM calculation, which retains only first-order terms, the parameter update already requires an additional forward and backward pass, undesirably doubling the computation time. Conversely, Equation (7) mitigates the computational burden and improves the precision of perturbations. □

**Remark 3.** *Loss landscape.* GNM can achieve a flat loss landscape for VPT.

Figure 1 and Section 4.4 empirically demonstrate the loss landscape obtained by GNM for VPT is flattened than the original VPT and SAM. Appendix I and Appendix L demonstrate that besides VPT, GNM can also improve other PEFT methods such as AdapterFormer [4] and other backbones such as ResNet based models.

Although GNM is not affected by class sizes and improves the generalization performance of each category equally, head-class bias caused by the classifier still exists. Two-stage strategy [56, 23, 10] is effective. Classifier re-balance strategy, including deferred re-weighting/sampling (DRW or DRS) [2], classifier Re-training (cRT) [23], nearest class mean classifier (NCM) [23], to name a few, can be employed. The overall training procedure of GNM-PT is summarized in Appendix D.

### 3.3 Theoretical Analysis of GNM

This section explains GNM from the theoretical perspective. We introduce the following theorem to demonstrate the compactness of the upper bound of the loss function across the distribution $\mathcal{D}$.

---

[3] The loss landscape for the background adheres to the same settings in Figure 2 of SAM [14].

**Theorem 1.** *For any $0 < \delta < 1$, and number of samples $n \in \mathbb{N}^+$, with probability $1 - \delta$ over the training set $\mathcal{T}$ sampled i.i.d. from a distribution $\mathcal{D}$, the following generalization bound w.r.t. model parameters $\boldsymbol{\theta}$ holds:*

$$L_{\mathcal{D}}(\boldsymbol{\theta}) \leq \mathbb{E}_{\varepsilon_i \in \mathcal{N}(0,\sigma^2)} \left[ L_{\mathcal{T}}(\boldsymbol{\theta} + \boldsymbol{\varepsilon}) \right] + h(\frac{\|\boldsymbol{\theta}\|_2^2}{4\sigma^2}), \tag{9}$$

*where $h : \mathbb{R}^+ \rightarrow \mathbb{R}^+$ is a strictly increasing function.*

*Proof.* Based on the condition that adding Gaussian perturbation should not decrease the test error, $L_{\mathcal{D}}$ satisfy:

$$L_{\mathcal{D}}(\boldsymbol{\theta}) \leq \mathbb{E}_{\varepsilon_i \in \mathcal{N}(0,\sigma^2)} \left[ L_{\mathcal{D}}(\boldsymbol{\theta} + \boldsymbol{\varepsilon}) \right]. \tag{10}$$

By Theorem 2 in Foret et al. [14] and Theorem 1 in Zhou et al. [71], the Gaussian perturbation satisfy:

$$\mathbb{E}_{\varepsilon_i \in \mathcal{N}(0,\sigma^2)} \left[ L_{\mathcal{D}}(\boldsymbol{\theta} + \boldsymbol{\varepsilon}) \right] \leq \mathbb{E}_{\varepsilon_i \in \mathcal{N}(0,\sigma^2)} \left[ L_{\mathcal{T}}(\boldsymbol{\theta} + \boldsymbol{\varepsilon}) \right]$$
$$+ \sqrt{\frac{\frac{1}{4}k \log\left(1 + \frac{\|\boldsymbol{\theta}\|_2^2}{k\sigma^4}\right) + \log(\frac{n}{\delta}) + 2\log(6n + 3k) + \frac{1}{4}}{n - 1}}, \tag{11}$$

where $k$ is the dimension of $\boldsymbol{\theta}$. Since $\log(1 + x) < x$ holds for all $x > 0$, Equation (11) can be simplified to:

$$\mathbb{E}_{\varepsilon_i \in \mathcal{N}(0,\sigma^2)} \left[ L_{\mathcal{D}}(\boldsymbol{\theta} + \boldsymbol{\varepsilon}) \right] \leq \mathbb{E}_{\varepsilon_i \in \mathcal{N}(0,\sigma^2)} L_{\mathcal{T}}(\boldsymbol{\theta} + \boldsymbol{\varepsilon})$$
$$+ \sqrt{\frac{\frac{\|\boldsymbol{\theta}\|_2^2}{4\sigma^2} + \log(\frac{n}{\delta}) + \mathcal{O}(1)}{n - 1}}. \tag{12}$$

The term containing square roots in the above expression is a strictly increasing function. Therefore, by combining it with Equation (10), Theorem 1 is proved. $\square$

Similar to [14, 71], $h$ in Equation (9) can be replaced by $L_2$ weight decay regularizer $\frac{\lambda}{2}\|\boldsymbol{\theta}\|_2^2$. Minimizing $L_{\mathcal{D}}$ can be achieved by solving the following GNM problem:

$$\min_{\boldsymbol{\theta}} L_{\mathcal{T}}^{GN}(\boldsymbol{\theta}) + \frac{\lambda}{2}\|\boldsymbol{\theta}\|_2^2. \tag{13}$$

Hence, the parameter updates given by Equation (7) and Equation (8) for GNM can minimize the upper bound given of $L_{\mathcal{D}}(\boldsymbol{\theta})$ by Theorem 1.

**Remark 4.** *Upper Bound for Loss Function.* GNM achieves a tighter upper bound for loss function than SAM.

*Proof.* According to Equation (2), $L_{\mathcal{T}}^{SAM}$ is obtained by minimizing the maximum of the loss within the parameter neighborhood of radius $\rho_{SAM}$. By adjusting the variance $\sigma$, $L_{\mathcal{T}}^{GN}$ is obtained by minimizing the average value of the loss function within the parameter neighborhood $r_{GNM}$. It is evident that when $\rho_{SAM} \geq \rho_{GNM}$, $\mathbb{E}_{\rho_{SAM}} \left[ L_{\mathcal{T}}(\boldsymbol{\theta} + \boldsymbol{\varepsilon}) \right] \leq \max_{\rho_{GNM}} \left[ L_{\mathcal{T}}(\boldsymbol{\theta} + \boldsymbol{\varepsilon}) \right]$. Therefore, $L_{\mathcal{T}}^{GN}$ is a tighter upper bound on the loss over $\mathcal{D}$ than $L_{\mathcal{T}}^{SAM}$. $\square$

## 4 Experiment

### 4.1 Datasets

**CIFAR100-LT.** We adopt the same settings utilized in [8, 2] to establish the long-tailed version by downsampling the original CIFAR100 dataset [25] with different imbalance ratios $IR = n_{\max}/n_{\min}$, where $n_{\max}, n_{\min}$ represent the class sizes of the most and the least frequent classes, respectively. Following [32], we set the imbalance ratios at 200, 100, 50, and 10.

**Places-LT.** It is artificially truncated from its balanced version, Places365 [67]. The long-tailed version was first created by Liu et al. [37]. Places-LT consists of 62.5K training images with an imbalance ratio of 996.

**iNaturalist2018.** iNaturalist is a substantial real-world dataset that inherently exhibits an exceedingly imbalanced distribution. In our experiments, we leverage the widely employed 2018 version [53] (iNat for short), encompassing 437.5K images across 8,142 distinct species. This dataset features an imbalance ratio of 512.

## 4.2 Implementation Details

**Evaluation Protocol.** Following the fundamental assumption that every class carries equal importance, all classes with varying frequencies in the training set are granted an equal number of samples during testing. Top-1 classification accuracy is the primary metric for assessing the performance of various methods. Following Liu et al. [37], we additionally provide accuracy measurements for three class splits based on the number of training data: Head ($> 100$ images), Medium (Med for short, $20 \sim 100$ images), and Tail ($\leq 20$ images).

**Model and Parameter Settings.** Following Dong et al. [10], we employ ViT-B/16 pre-trained on ImageNet-21K and VPT-deep for prompt tuning, and GCL [32] as the loss function. The same data augmentation strategies outlined in [48, 29, 22] are adopted, consistent with widely adopted practices among mainstream methods. We employ SGD with GNM as an optimizer and set the batch size to 128, a learning rate of 0.01, accompanied by a cosine learning rate scheduler. For parameter settings of the Gaussian distribution parameters $(0, \sigma^2)$ mentioned in Section 3.2, we exploit the same strategy as Li et al. [32]. Specifically, we set $\sigma = \frac{1}{3}$ meanwhile clamping $\varepsilon$ within the range $[-1, 1]$ to ensure that its amplitude remains within one and use a hyper-parameter $\rho$ to control the perturbation magnitude. We adopt DRW for classifier re-balance. Notably, LPT [10] also employs a re-balance strategy during the group prompt tuning stage. For CIFAR100-LT and iNat, we fine-tuned models for 70 epochs, with the final 10 epochs for DRW. For Places-LT, the models undergo a fine-tuning process spanning 100 epochs, with the last 40 epochs for DRW.

## 4.3 Comparison with Prior Arts

### 4.3.1 Compared Methods

We compare our proposed GNM-PT with several state-of-the-art methods, broadly categorized into two main types.

Table 1: Comparison on CIFAR100-LT w.r.t top-1 classification accuracy (%).

| Method | 200 | 100 | 50 | 10 |
|---|---|---|---|---|
| DNN-based model (Backbone: ResNet32) | | | | |
| BBN [68] | 37.2 | 42.6 | 47.0 | 59.1 |
| RIDE [57] | 45.8 | 50.4 | 55.0 | - |
| MiSLAS [66] | 43.5 | 47.0 | 52.3 | 63.2 |
| BCL [72] | - | 51.9 | 56.6 | 64.9 |
| GCL [32] | 44.8 | 48.6 | 53.6 | - |
| NCL [29] | - | 54.2 | 58.2 | - |
| GPaCo [7] | - | 52.3 | 56.4 | 65.4 |
| SHIKE [22] | - | 56.3 | 59.8 | - |
| DNN-based model with SAM | | | | |
| CCSAM [71] | 45.7 | 50.8 | 53.9 | - |
| ImbSAM [70] | - | 54.8 | 59.3 | 59.7 |
| Self-attention-based model (Backbone: ViT-B/16) | | | | |
| VPT [21] | 72.8 | 81.0 | 84.8 | 89.6 |
| LiVT [62] | - | 58.2 | - | 69.2 |
| LPT [10] | **87.9** | **89.1** | **90.0** | **91.0** |
| GNM-PT (ours) | **_89.2_** | **_90.3_** | **_91.2_** | **_91.8_** |

**Note**: The best and second-best results are shown in **_underline bold_** and **bold**, respectively.

**DNN-based model.** We compare with (1) two-stage methods, namely BBN [68], LWS [23], MiSLAS [66]; (2) logit adjustment methods, i.e., GCL [32] and LDAM [2]; (3) ensembling learning methods, including, RIDE [57], NCL [29], and SHIKE [22]; and (4) contrastive learning, represented by GPaCo [7]. Additionally, we compared two recently proposed SAM-based methods, CCSAM [71] and ImbSAM [70], explicitly designed to address long-tail data.

**MHSA-based model.**

Recently, MHSA-based models represented by ViT have been employed in long-tail visual recognition. We compare with visual-only methods, including LiVT [62], VPT [21], BALLAD [39], LPT [10], and Decoder [60]. All methods were implemented using ViT-B/16 for a fair comparison. In addition, VL-LTR [52], RAC [38] and GML [51] are also MHSA-based models which use supplementary data (i.e., text information). We also report the results obtained by these methods for reference.

### 4.3.2 Comparison Results

**Comparison on CIFAR100-LT.** We present the comparison results for CIFAR100-LT in Table 1. Our proposed GNM-PT exhibits superior performance across all commonly used imbalance ratios compared to the competing methods. Notably, as the imbalance ratio increases, the effectiveness of our GNM-PT becomes increasingly apparent on CIFAR100-LT. Specifically, our proposed method

achieves improvements of 1.3%, 1.2%, 1.2%, and 0.8% over the second-best method, namely LPT [10], for imbalance ratios of 200, 100, 50, and 10, respectively.

**Comparison on iNat.** Table 2 provides results on iNat. The proposed GNM-PT achieves a top-1 classification accuracy of 76.5%, surpassing DNN-based methods. Compared with other visual-only MHSA-based methods that exclusively rely on visual data, our improvement may not be substantial (76.5% However, it is noteworthy that GNM-PT is trained with a relatively small number of epochs (70 and 80 epochs without and with DRW, respectively). In contrast, LPT [10] requires 160 epochs (80 for shared prompt tuning and 80 for group prompt tuning), while LiVT [62] requires 100 epochs. Our proposed method can even achieve results comparable to those with supplementary data. For example, VL-LTR [52], which requires image-text pairs, achieves an accuracy of 76.8%, only 0.3% and 0.5% higher than GNM-PT with and without DRW, respectively. Notably, the adoption of DRW in GNM-PT has the potential to enhance overall performance, albeit with the trade-off of sacrificing head-class accuracy to bolster tail-class accuracy. This observation may be attributed to suboptimal parameter selection in calculating effective numbers in DRW, an aspect that we plan to delve into in future work.

Table 2: Acc. (%) comparison on iNat.

| Method | Head | Med | Tail | Overall |
|---|---|---|---|---|
| DNN-based model (Backbone: ResNet50) | | | | |
| LWS [23] | 72.9 | 71.2 | 69.2 | 70.5 |
| RIDE [57] | 76.5 | 74.2 | 70.5 | 72.8 |
| MisLAS [66] | 73.2 | 72.4 | 70.4 | 71.6 |
| GCL [32] | - | - | - | 72.0 |
| NCL [29] | 72.7 | 75.6 | 74.5 | 74.9 |
| GPaCo [7] | - | - | - | 75.4 |
| SHIKE [22] | - | - | - | 75.4 |
| DNN-based model with SAM | | | | |
| LDAM+SAM [47] | 64.1 | 70.5 | 71.2 | 70.1 |
| CCSAM [71] | 65.4 | 70.9 | 72.2 | 70.9 |
| ImbSAM [70] | 68.2 | 72.5 | 72.9 | 71.1 |
| MHSA-based model (Backbone: ViT-B/16) | | | | |
| Supplementary with linguistic data | | | | |
| VL-LTR [52] | - | - | - | **76.8**[3] |
| RAC [38] | 75.9 | 80.5 | 81.1 | **80.2**[3] |
| Visual-only | | | | |
| Decoder [60] | - | - | - | 59.2 |
| LPT [10] | - | - | 79.3 | 76.1 |
| LiVT [62] | 78.9 | 76.5 | 74.8 | 76.1 |
| GNM-PT (ours) | 61.5 | 77.1 | 79.3 | **76.5** |
| GNM-PT (ours) | 76.3 | 77.6 | 75.0 | **76.3**[4] |

Table 3: Acc. (%) comparison on Places-LT.

| Method | Head | Med | Tail | Overall |
|---|---|---|---|---|
| DNN-based model (Backbone: ResNet152) | | | | |
| LWS [23] | 40.6 | 39.1 | 28.6 | 37.6 |
| RIDE [57] | 44.4 | 40.6 | 33.0 | 40.4 |
| MisLAS [66] | 39.6 | 43.3 | 36.1 | 40.4 |
| GCL [32] | 38.6 | 42.6 | 38.4 | 40.3 |
| NCL [29] | - | - | - | 41.8 |
| GPaCo [7] | 39.5 | 47.2 | 33.0 | 41.7 |
| SHIKE [22] | 43.6 | 39.2 | 44.8 | 41.9 |
| DNN-based model with SAM | | | | |
| CCSAM [71] | 41.2 | 42.1 | 36.4 | 40.6 |
| MHSA-based model (Backbone: ViT-B/16) | | | | |
| Supplementary with linguistic data | | | | |
| VL-LTR [52] | 54.2 | 48.5 | 42.0 | **50.1**[3] |
| RAC [38] | 48.7 | 48.3 | 41.8 | 47.2[3] |
| Visual-only | | | | |
| Decoder [60] | - | - | - | 46.8 |
| LPT [10] | 47.6 | 52.1 | 48.4 | 49.7[5] |
| LiVT [62] | 48.1 | 40.6 | 27.5 | 40.8 |
| GNM-PT (ours) | 46.6 | 53.3 | 49.4 | **50.1** |
| GNM-PT (ours) | 48.6 | 52.1 | 47.9 | **50.0**[4] |

**Comparison on Places-LT.** From Table 3, we can observe that GNM-PT continues to outperform existing methods. Similarly to iNat, GNM-PT obtains performance equivalent to LPT with fewer training epochs and outperforms LiVT by nearly 10%. Even when compared to VL-LTR and RAC, which leverage additional auxiliary data, GNM-PT still achieves satisfying performance. Additionally, from Table 3, it can be observed that DRW improves tail classes at the expense of significant degradation in head classes on Places-LT. While it resulted in an overall improvement of 1%, the head classes decreased by 2%. This indicates that the chosen effective number employed by DRW may not be optimal, warranting further investigation. More results on imageNet-LT can be found in Appendix G.

---

[3]The results are obtained with the assistance of textual data.

[4]The result is obtained without DRW.

[5]The results are obtained by reproducing the original paper with the recommended settings.

## 4.4 Further Analysis

To ensure a fair comparison, the experiments in this section are all executed on the following hardware: Core(TM) i9-13900K, operating at 3.00GHz, equipped with 128GB RAM, and a single NVIDIA GeForce RTX 4090 GPU. The dataset is CIFAR100-LT with $IR = 100$.

**GNM vs. SAM.** To demonstrate the superiority of GNM, we conducted a comparative analysis with the SAM from two perspectives: classification accuracy and computational efficiency. We employ both CE loss and GCL loss utilizing the CIFAR100-LT dataset with an imbalance ratio of 100. Except for incorporating the optimization of SAM or GNM, all other settings remain identical. Table 4 shows the results. We can observe that SAM entails a computation time exceeding 1.8 times that of the original method compared to the baseline methods without additional optimization technologies. In contrast, GNM incurs only a negligible increase in computation time, namely, less than 2 seconds per epoch. In addition to time savings, GNM also manifests a discernible improvement in accuracy. Despite our straightforward adoption of random perturbation vector $\tilde{\varepsilon}$ (as detailed in Section 3.2), its performance is superior to that of the theoretical optimal perturbation vector $\hat{\varepsilon}$ for seeking the maximum loss value within the neighborhood. It is worth noting that all experiments in this section employ the same number of training epochs. SAM and GNM *do not* affect the convergence speed of the network. The effectiveness across various classes is visualized in Figure 3. While SAM declines the performance of GCL within the tail classes, our proposed GNM consistently improves performance across all categories in every scenario. Additional comparison results for the long-tailed SAM method can be found in Appendix J and Appendix K. Further comparisons for balanced softmax (BASM) [48] and ResNet-152 backbone can be found in Appendix L

Table 4: Comparison between SAM and the proposed GNM. NET represents Native Execution Time.

| Method | Acc. (%) | NET (s) |
|---|---|---|
| CE | 81.02 | 39.78 |
| CE+SAM | 82.48 | 72.51 |
| CE+GNM | **82.50** | 40.16 ($\downarrow$ 44.61% ) |
| GCL+DRW | 89.58 | 40.00 |
| GCL+DRW+SAM | 89.69 | 74.36 |
| GCL+DRW+GNM | **90.28** | 41.87 ($\downarrow$ 43.69% ) |

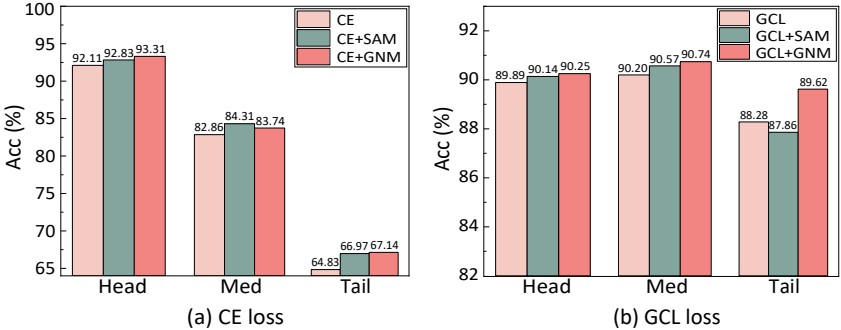

Figure 3: Effectiveness comparison of different classes.

**Visualization of Loss Landscape.** We employ the method in [28] to visualize the loss landscape of the model. Figure 4 shows the results of the learnable prompts obtained by different optimizers. The absence of a perceptible change in flatness explains the marginal improvement of GCL+DRW+SAM over GCL+DRW in Table 4. In comparison, GNM, by inducing a flattened loss landscape, further accentuates the improvement over GCL+DRW. An unforeseen advantage is that GNM results in a smaller loss, indicating that GNM enhances the model fitting to the training data. A flatter landscape with lower minima contributes to discovering a more optimal solution. By referring to Figure 1 in Section 1, it can be observed that GNM is effective across various loss functions.

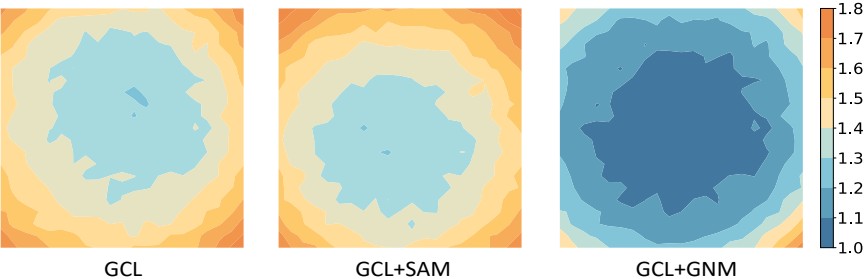

Figure 4: GCL loss landscapes based on ViT-B/16 (best view in color).

## 5 Concluding Remarks

In this paper, we observed that class biases persist even when employing large-scale pre-trained models such as VPT in long-tail learning. While SAM can enhance the generalization performance of the VPT model on long-tailed data, it still exhibits several shortcomings: neglecting higher-order terms leads to a suboptimal perturbation vector, additional forward and backward passes double the computational time, and the generalization is relatively affected by gradients predominantly originating from head classes. Based on this, we have proposed GNM-PT, which involves fine-tuning pre-trained models using the innovative Gaussian Neighborhood Minimization (GNM) optimizer. GNM leverages random noise as a substitute for gradients in the initial step of SAM. The proposed GNM not only balances the generalization capabilities of both head and tail classes but also reduces computational time. To fully leverage model information, enhance classifier robustness, and enable end-to-end training, we additionally employ merging prompt strategy. We have conducted extensive comparative experiments and ablation studies to demonstrate the effectiveness of the proposed method and the individual component.

While GNM-PT proves effective, it is not exempt from limitations. Table 2 and Table 3 show that we still need to further re-balance the classifier. However, the re-balanced strategy adopted compromises performance in head classes to enhance overall performance. Our further research will focus on a more effective optimization strategy that simultaneously improves feature representation and classifier performance, while also enhancing the generalization ability across all classes.

## Acknowledgement

We thank the reviewers for their valuable comments. This work was supported in parts by NSFC (62306181, U21B2023, 62476063, 62431004), Guangdong Basic and Applied Basic Research Foundation (2024A1515010163, 2023B1515120026), Shenzhen Science and Technology Program (RCBS20231211090659101, KQTD20210811090044003, RCJC20200714114435012), NSFC/RGC (N_HKBU214/21), RGC GRF (12201323), and Research Funds from Shenzhen University.

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

# Appendix

## A  Gradient Computation for GNM

We replace the expectation computation in $L_{\mathcal{T}}^{GN}$ with random sampling during training. For a single gradient computation, we have:

$$\nabla_{\boldsymbol{\theta}} L_{\mathcal{T}}^{GN}(\boldsymbol{\theta}) \approx \nabla_{\boldsymbol{\theta}} L_{\mathcal{T}}(\boldsymbol{\theta} + \tilde{\boldsymbol{\varepsilon}}) \tag{14}$$

$$= \frac{d(\boldsymbol{\theta} + \tilde{\boldsymbol{\varepsilon}})}{d\boldsymbol{\theta}} \nabla_{\boldsymbol{\theta}} L_{\mathcal{T}}(\boldsymbol{\theta})|_{\boldsymbol{\theta} + \tilde{\boldsymbol{\varepsilon}}} = \nabla_{\boldsymbol{\theta}} L_{\mathcal{T}}(\boldsymbol{\theta})|_{\boldsymbol{\theta} + \tilde{\boldsymbol{\varepsilon}}}. \tag{15}$$

For simplicity, we omit the subscript indicating the training epoch $t$.

## B  Detail Proof of Remark 1

If $\mathcal{T}$ is a long-tail training set i.i.d. sampled from $\mathcal{D}$, the direction of gradients in existing methods such as SGD and SAM is predominantly influenced by head classes. For the most widely adopted SGD, a lot of previous works, such as Wang et al. [55], Hsieh et al. [19], Li et al. [31] have theoretically and empirically demonstrated that the gradient for head classes far exceeds that of tail classes. Therefore, the optimization is dominated by head classes. Here, we provide proof establishing that the perturbations within SAM are dominated by head classes.

*Proof.* For SAM, we analyze the perturbations class by class. Through Eq. (4), we have:

$$\boldsymbol{\varepsilon} \leftarrow \hat{\boldsymbol{\varepsilon}} = \rho \frac{\nabla_{\boldsymbol{\theta}} L_{\mathcal{T}^{\mathrm{Tail}}}(\boldsymbol{\theta}) + \nabla_{\boldsymbol{\theta}} L_{\mathcal{T}^{\mathrm{Head}}}(\boldsymbol{\theta})}{\|\nabla_{\boldsymbol{\theta}} L_{\mathcal{T}}(\boldsymbol{\theta})\|_2^2}. \tag{16}$$

The $\rho \frac{1}{\|\nabla_{\boldsymbol{\theta}} L_{\mathcal{T}}(\boldsymbol{\theta})\|_2^2}$ can be considered to be the same value $V$ for both the head and tail classes. Wang et al. [55], Li et al. [31] have theoretically and empirically shown that the gradient for head classes far exceeds that of tail classes, that is, $\nabla_{\boldsymbol{\theta}} L_{\mathcal{T}^{\mathrm{Tail}}} \gg \nabla_{\boldsymbol{\theta}} L_{\mathcal{T}^{\mathrm{Tail}}}$. Therefore, the perturbation obtained based on tail classes can be ignored:

$$\boldsymbol{\varepsilon} \approx \rho \frac{\nabla_{\boldsymbol{\theta}} L_{\mathcal{T}^{\mathrm{Head}}}(\boldsymbol{\theta})}{\|\nabla_{\boldsymbol{\theta}} L_{\mathcal{T}}(\boldsymbol{\theta})\|_2^2} = V \cdot \nabla_{\boldsymbol{\theta}} L_{\mathcal{T}^{\mathrm{Head}}}(\boldsymbol{\theta}). \tag{17}$$

Consequently, SAM optimization tends to prioritize generalization for head classes.  $\square$

In contrast, the perturbations obtained by GNM are:

$$\boldsymbol{\varepsilon} \leftarrow \tilde{\boldsymbol{\varepsilon}}_t = [\varepsilon_i]_{i=1}^k , \ \varepsilon_i \sim \mathcal{N}(0, \sigma^2) \tag{18}$$

This perturbation remains unaffected by the input samples and their quantities.

## C  Robust Training strategy for Prompt Tuning.

In Eq. (1), only $\mathbf{z}_{cls}^l$ is fed into the linear classifier for classification. However, all the learnable prompts are trained on the fine-tuning dataset and thus contain newly learned information. As the deep block output incorporates global attention [11], we propose merging prompt (MP) that merges the last prompt with $\mathbf{z}_{cls}^L$ (assuming that we have $L$ blocks). Subsequently, we utilize this merged token as the ultimate class token:

$$\hat{\mathbf{z}}_{cls} = \mathtt{Merge}\left(\left[w_p \cdot \mathbf{p}^{L-1}, w_z \cdot \mathbf{z}_{cls}^L\right]\right), \tag{19}$$

where $w_p$ and $w_z$ are hyper-parameters used to control the merging ratio. $\hat{z}_{cls}$ is eventually fed into the linear classifier for classification. We use this Merge Prompt technique in our experiments.

## D  Algorithm for GNM-PT

The training procedure for GNM-PT is outlined in Algorithm 1.

**Algorithm 1** GNM-PT
___
**Input:** Training set $\mathcal{T}$, pre-trained model
**Output:** Fine-tuned model
Initialize the prompt randomly with parameters $\theta$
**for** $t = 1$ **to** $T_1$ **do** {robust training}
    Sample batches data $\mathcal{B}_t \sim \mathcal{T}$
    Compute perturbation $\tilde{\varepsilon}_t \rightarrow \varepsilon_t$ by Eq. (7)
    Obtain class token $\hat{z}_{cls}$ by Equations (1) and (19)
    Compute GNM gradient: $g_t = \nabla_{\boldsymbol{\theta}} \dfrac{1}{|\mathcal{B}_t|} L_{\mathcal{B}_t}(\hat{z}_{cls}, \boldsymbol{\theta}_t) \mid_{\theta + \varepsilon_t}$
    Update fine-tuning parameters: $\boldsymbol{\theta}_{t+1} = \boldsymbol{\theta}_t - \alpha_t \cdot g_t$
**end for**
**for** $t = T_1 + 1$ **to** $T_2$ **do** {re-balance classifier}
    Sample batches data $\mathcal{B}_t \sim \mathcal{T}$
    Compute perturbation $\tilde{\varepsilon}_t \rightarrow \varepsilon_t$ by Eq. (7)
    Obtain class token $\hat{z}_{cls}$ by Equations (1) and (19)
    Compute re-weight parameter $w_c$ for class $c$
    Compute GNM gradient: $g_t = \nabla_{\boldsymbol{\theta}} \dfrac{1}{|\mathcal{B}_t|} \sum_{c=1}^{C} w_c L_{\mathcal{B}_t}^c(\hat{z}_{cls}, \boldsymbol{\theta}_t) \mid_{\boldsymbol{\theta}_t + \boldsymbol{\varepsilon}_t}$
    Update fine-tuning parameters.
**end for**
___

# E    Schematic Comparison of SAM and GNM

Figure 5 compares the optimization directions of SAM and GNM. SAM achieves the flattening of the loss landscape by introducing perturbations in the opposite direction of gradient descent. In contrast, GNM accomplishes the flattening of the loss landscape by introducing random perturbations in the parameter neighborhood using the Monte Carlo method.

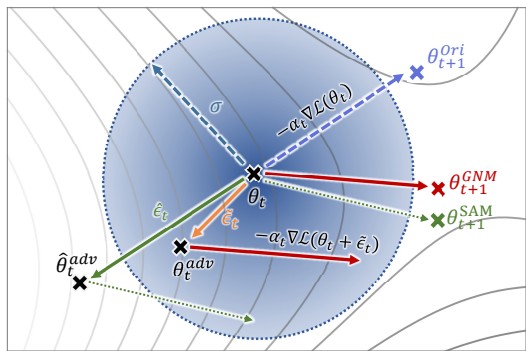

Figure 5: Comparison of optimization directions. $\boldsymbol{\theta}_{t+1}^{Ori}$, $\boldsymbol{\theta}_{t+1}^{SAM}$ and $\boldsymbol{\theta}_{t+1}^{GNM}$ represent the original gradient update, gradient update with SAM and with GNM for step $t+1$, respectively.

# F    Ablation Study for GNM and Merge Prompt

To validate the effectiveness of each proposed component, we conducted an ablation study using the CIFAR100-LT dataset with an imbalance ratio of 100, employing GCL loss. For "Merge" in Eq. (19), we employ addition and assign equal weights to both $w_p$ and $w_z$, setting them to 0.5. The summarized results are presented in Table 5. Incorporating DRW has been observed to enhance overall performance, resulting in a performance improvement of 1.07%. Additionally, GNM-PT derives benefits from the design choices made in each individual component. The proposed merge prompt and GNM optimization further improve the performance from 89.32% to 89.58% and 90.28%, respectively.

Table 5: Effect of each component in the proposed GNM-PT on CIFAR-100-LT with imbalance ratio = 100.

| DRW | Merge Prompt | GNM | Acc. (%) |
|---|---|---|---|
| ✗ | ✗ | ✗ | 88.25 |
| ✓ | ✗ | ✗ | 89.32 |
| ✓ | ✓ | ✗ | 89.58 |
| ✓ | ✓ | ✓ | **90.28** |

# G   Comparison on ImageNet-LT

Similar to Place-LT, imageNet-LT is also artificially truncated from its balanced version, namely ImageNet [9] and its long-tailed version is created by Liu et al. [37]. ImageNet-LT comprises 115.8K training images spanning across 1,000 categories, with an imbalance ratio of 256.

We report the accuracy in Table 6. Considering that the model pre-trained on ImageNet-21K contains information about ImageNet-1K, namely the balanced version of ImageNet-LT, we compare GNM-PT only with the methods using the same pre-trained models. GNM-PT achieves superior performance, attaining an 80.4% top-1 classification accuracy, outperforming GML and VPT with a notable margin of 2.4% and 3.2%, respectively. Furthermore, GNM-PT surpasses competing methods across all scale classes, demonstrating its outstanding performance.

Table 6: Comparison results on ImageNet-LT.

| Method | Head | Med | Tail | Overall |
|---|---|---|---|---|
| Backbone: ViT-B/16 | | | | |
| Supplementary with linguistic data | | | | |
| VL-LTR [52] | 84.5 | 74.6 | 59.3 | 77.2[3] |
| GML [51] | - | - | - | **78.0**[3] |
| Visual-only | | | | |
| BALLAD [39] | 79.1 | 74.5 | 69.8 | 75.7 |
| VPT [21] | 79.5 | 76.5 | 72.8 | **77.2** |
| Decoder [60] | - | - | - | 73.2 |
| GNM-PT | 80.6 | 81.1 | 78.2 | **80.4** |

# H   Ablation Study for Hyperparameters

In the comparison experiment with SAM, we used the radius ($r_{SAM}$) recommended by the paper in SAM, which is 0.05. For GNM, we set the amplitude ($a$) for Gaussian noise based on the radius in SAM, that is, the actual radius ($r_{GNM}$) used in GNM is $\rho_{GNM} = a \times \rho_{SAM}$. We use $a = 0.1$ in all experiments. We conducted the ablation study towards the hyper-parameter $a$ in GNM using CIFAR100 with an imbalance ratio of 100. The results are listed in Table 7. When $\alpha \to 0$, the interference becomes negligible, effectively restricting the loss function to attain its minimum value within a small area. The extreme case is $\alpha = 0$, meaning no additional optimization techniques are employed. Therefore, the smaller $\alpha$ is, the less pronounced its effect. When $\alpha$ increases: the disturbance area expands. A large $\alpha$ introduces significant perturbations, potentially deviating from the basic gradient descent path. Excessively large values of $\alpha$, for example $\alpha = 2$, lead to performance degradation. In the extreme case of $\alpha = \infty$, the model fails to converge.

Table 8 presents ablation studies on the different choices of the variance of the Gaussian distribution. Additionally, we include the results of using a uniform distribution within the range of [-1,1]. The results indicate that the variance impacts model performance. This finding demonstrates that, in addition to the amplitude of $\tilde{\epsilon}$, the distribution also influences model performance and is worthy of further study.

---

[3]The results are obtained with the assistance of textual data.

Table 7: Ablation study for $a$ on CIFAR-100-LT with imbalance ratio $= 100$.

| $a$ | 0.01 | 0.05 | 0.1 | 0.2 | 0.5 | 1.0 | 2.0 | SAM ($\rho_{SAM} = 0.05$) |
|---|---|---|---|---|---|---|---|---|
| Acc. (%) | 90.03 | 90.31 | 90.28 | 90.05 | 89.54 | 89.71 | 88.45 | 89.69 |

Table 8: Ablation study for variance on CIFAR-100-LT with imbalance ratio $= 100$.

| $\rho$ | 3 | 2 | 1 | 0.8 | 0.6 | 0.4 | (1/3) | 0.2 | (uniform distribution) |
|---|---|---|---|---|---|---|---|---|---|
| Acc. (%) | 90.03 | 89.99 | 90.14 | 90.23 | 90.01 | 90.06 | (90.28) | 90.12 | (90.17) |

# I  Experimental Results for Applying GNM on AdapterFormer

AdapterFormer [4] is one of the recently proposed PEFT techniques. We show the efficacy of GNM when applied to AdapterFormer. The results are summarized in Table 9. In contrast to prompt tuning-based approaches, GNM demonstrates relatively modest improvements on AdapterFormer. The reason is that, compared to prompt tuning methods, AdapterFormer has fewer learnable parameters, comprising 1.01M and 0.18M parameters, respectively.

Table 9: Results for AdaptFormer with different pre-trained ViT on CIFAR-100-LT with imbalance ratio $= 100$. IN21K is short for ViT pre-trained with imageNet-21K. CLIP is introduced by Radford et al. [46].

| Method | IN21K-SGD | IN21K-SAM | IN21K-GNM | CLIP-SGD | CLIP-SAM | CLIP-GNM |
|---|---|---|---|---|---|---|
| Acc. (%) | 89.14 | 89.07 | **89.26** | 81.70 | 81.88 | **81.96** |

# J  Comparison with SAM-based Method

Table 10 and Table 11 compare the SAM-based method for long-tailed data, that is, CC-SAM [71]. GNM consistently enhances generalization performance across each class compared to CC-SAM.

Overall, GNM can serve as an optimization method that not only enhances the performance of the VPT-based model but also improves the performance of CNNs trained from scratch or with full fine-tuning.

Table 10: Comparison results on CIFAR-100-LT. The backbone is ResNet32.

| Imbalance ratio | 200 | 100 | 50 |
|---|---|---|---|
| CC-SAM | 45.66 | 50.83 | 53.91 |
| GNM | 46.33 | 51.13 | 54.50 |

Table 11: Comparison results on Places-LT. The backbone is ResNet152.

| Method | Head | Med | Tail | All |
|---|---|---|---|---|
| CC-SAM | 43.69 | 41.95 | 31.95 | 40.46 |
| GNM | 43.92 | 42.13 | 32.74 | 40.79 |

# K  Comparison Results w.r.t. Optimization Strategies Under the Same Backbone

Tables 12 and 13 present the comparison of existing optimization strategies and our GNM under the same training paradigm. We implement the experiments using CIFAR-100-LT with an imbalance ratio of 100 and take fine-tuning pre-trained ViT-B/16 by VPT with GCL loss as a base training paradigm, named Base. The re-balancing strategy employed in the second stage can influence the performance of optimization methods on a per-class level. We conducted a comparative analysis for various optimization methods, both without and with the application of the rebalancing strategy and exhibit the result in Tables 12 and 13, respectively. DRW is utilized as the re-balancing strategy. LPT also employs two stages that include a re-balance strategy, therefore we present it in Table 13.

As observed from Table 12, without the re-balancing strategy to adjust the classifier bias, imbSAM achieves better overall accuracy. However, ImbSAM has little impact on the head and middle classes. Additionally, both CCSAM and ImbSAM require two back propagations, thereby doubling the computation time. Compared to VPT with GCL, which does not include additional optimization, GNM incurs only a small computational overhead.

Table 13 shows that the re-balancing strategy sacrifices a small amount of head class performance in exchange for significantly improving tail class performance. imbSAM essentially does not employ additional optimization for head classes, whereas CCSAM and GNM use additional optimization for all classes. However, compared to imbSAM, CCSAM and GNM result in a greater reduction in the accuracy of head classes. The optimization strategies may have a limited impact on the rebalancing training process. We will investigate this in detail in future work.

Besides, we provide a detailed analysis of why GNM-PT cannot outperform imbSAM in the first stage but outperform in the second stage as shown in Table 12 and Table 13. Stage 1 of ImbSAM already applies strong regularization to tail classes, the rebalancing strategy in stage 2 essentially duplicates this effect. In contrast, GNM applies the same level of regularization to all classes without specifically intensifying it for tail classes and thus the strong regularization for tail classes can still work in stage 2.

Table 12: Optimization strategy comparison. Models are trained with stage 1 only. NET represents native execution time (s).

| Method | Head | Med | Tail | All | NET |
|---|---|---|---|---|---|
| Base | 92.86 | 88.94 | 79.28 | 87.76 | **40.32** |
| Base + CCSAM | 92.81 | 88.31 | 79.31 | 87.84 | 80.47 |
| Base + ImbSAM | 92.92 | 88.43 | **84.00** | **89.02** | 88.97 |
| Base + GNM-PT | **93.67** | **89.03** | 81.10 | 88.46 | 42.77 |

Table 13: Optimization strategy comparison (with stage 2). The listed decreases in accuracy (%) of head classes are compared to that in Table 12.

| Method | Head | Med | Tail | All |
|---|---|---|---|---|
| LPT | - | - | - | 89.10 |
| Base | 90.08 ($\downarrow$ 2.78) | 89.60 | 88.14 | 89.40 |
| Base + CCSAM | 90.47 ($\downarrow$ 2.34) | 89.63 | 88.03 | 89.54 |
| Base + ImbSAM | 91.75 ($\downarrow$ 1.17) | 88.71 | 87.90 | 89.62 |
| Base + GNM-PT | **91.94** | **90.17** | **88.21** | **90.28** |

## L   Experimental Results for Applying GNM with balanced softmax

Table 14 presents the results of applying GNM with balanced softmax (BASM) [48]. The backbone is ResNet-152 pretrained on imageNet-1k. GNM exhibits improvements across all classes, especially tail classes. Consequently, the overall performance is enhanced in comparison to SAM.

Figure 6 presents the visualization results for the places-LT trained with Resnet-152. Although, the performance differences are small as shown in Table 14, it can still be observed that SGD and SAM exhibit steeper gradients and some irregular protrusions.

Table 14: Comparison results on Places-LT with balanced softmax (BASM).

| Method | Head | Med | Tail | All |
|---|---|---|---|---|
| BASM-SGD | 43.71 | 42.66 | 27.05 | 39.75 |
| BASM-SAM | 43.79 | 42.85 | 27.31 | 39.91 |
| BASM-GNM | 43.93 | 41.96 | 30.57 | 40.27 |

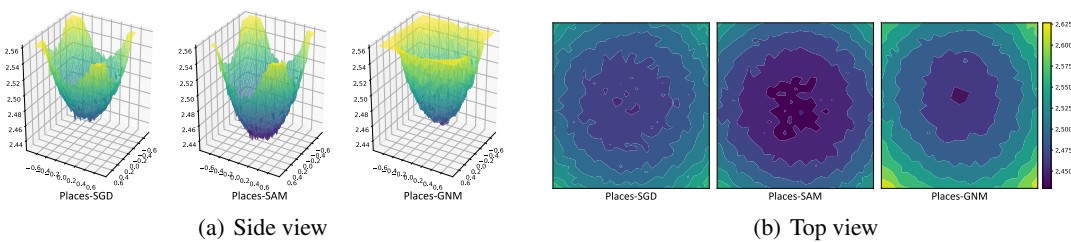

(a) Side view                     (b) Top view

Figure 6: Loss landscape comparison of ResNet-152 with BASM [48] (best view in color). The dataset used is Places-LT.

