# OpenReview forum: "Improving Visual Prompt Tuning by Gaussian Neighborhood Minimization for Long-Tailed Visual Recognition"
_NeurIPS.cc/2024/Conference — NeurIPS 2024 poster_

### Official Review · Reviewer_XQK2 · 2024-06-29

**Soundness:** 2
**Presentation:** 3
**Contribution:** 2
**Rating:** 5
**Confidence:** 4

**Summary:**

The author proposes a novel optimization strategy, Gaussian neighborhood minimization prompt tuning (GNM-PT), for VPT under long-tailed distribution. This method is improved based on the SAM optimizer. Specifically, during each gradient update, GNM-PT searches for a gradient descent direction within a random parameter neighborhood, independent of the input sample, which eliminates the influence of imbalance. Further experiments show the effectiveness of the method.

**Strengths:**

1) The proposed method is simple and effective.
2) The proposed method can improve the performance and accelerate the training procedure compared with SAM.
3) Solid theoretical supports.
4) The paper is well written.

**Weaknesses:**

1) It is not clear why the author applied GNM to VPT and not to adapterformer or lora or any other PEFT-based method or even training from scratch under CNN based model. All of these also face the problem of local minimum.
2) How to optimize under long-tailed distributions has been explored [a][b]. The authors should experiment with them directly under VPT and compare with GNM-PT and discuss their performance and training costs.
3) I admit that the flat minimum is mainly for the head class, but can we apply loss reweighting or logit adjustment[c] to alleviate this?
4) The author has done the experiment with the hyper-parameters in Appendix.G, but does not give any analysis of how this parameter influences the performance.

[a] Class-conditional sharpness-aware minimization for deep long-tailed recognition, CVPR

[b] A closer look at sharpness-aware minimization in class-imbalanced recognition, ICCV

[c] Long-tail learning via logit adjustment, ICLR2021

**Questions:**

see weakness

**Limitations:**

yes

---

> ### Author Rebuttal · Authors · 2024-08-06
>
> > Q1. Why apply GNM to VPT?
>
> **A1:** We acknowledge that most deep models encounter the issue of sharp local minima. We select VPT as a representative method. The primary reason is that it facilitates direct comparison with existing methods tailored for long-tailed learning, which employ the same fine-tuning strategy.
>
> For other training paradigms, we present the results obtained using AdapterFormer, fine-tuned on ImageNet21K and CLIP (refer to Table 8), as well as those from training the model from scratch (refer to Tables 9 and 10). These results demonstrate that GNM can further enhance the original models. We will include additional baseline models in our revised version.
>
> ---
> > Q2. Compare with CC-SAM and ImbSAM under VPT.
>
> **A2:** Thank you for this valuable comment. We implement the experiments using CIFAR-100-LT with an imbalance ratio of 100. The re-balancing strategy employed in the second stage can influence the performance of optimization methods on a per-class level, we conducted a comparative analysis for various optimization methods, both without and with the application of the rebalancing strategy. DRW is utilized as the re-balancing strategy. LPT also employs a two-stage strategy that includes a re-balance strategy, thus we present it in Table R6. The training cost of Stage 2 is similar to that of Stage 1; therefore, it is not listed again.
>
> **Table R5.** Optimization strategy comparison (w.o. stage2).
> |   Method                 |  Head acc. (%) | Med. acc. (%) | Tail acc. (%) | All acc. (%) |NET (s)
> | --------                 |-------| ----- |------ | -----|-----|
> |VPT (ViT-B/16) w. GCL     |  92.86 | *88.94*  |79.28  | 87.76 |**40.32**|
> |VPT (ViT-B/16) w. GCL and CCSAM |  92.81 | 88.31  |79.31  | 87.84 |80.47|
> |VPT (ViT-B/16) w. GCL and imbSAM|  *92.92* | 88.43  |**84.00**  | **89.02** | 88.97|
> |GNM-PT                    |  **93.39** | **89.14**  |*81.38*  | *88.58* | *42.77*|
>
> **Note:** NET represents native execution time.
>
>
> **Table R6.** Optimization strategy comparison (w. stage2).
> |   Method       |  Head acc. (%) | Med. acc. (%) | Tail acc. (%) | All acc. (%) |
> | --------       |-------| ----- |------ | -----|
> | LPT            | *     |   *   |     * |89.10 |
> |VPT (ViT-B/16) w. GCL  |90.08 ($\downarrow$ 2.78)|89.60|*88.14* | 89.40 |
> |VPT (ViT-B/16) w. GCL & CCSAM | 90.47 ($\downarrow$ 2.34)| *89.63* | 88.03 | 89.54 |
> |VPT (ViT-B/16) w. GCL & imbSAM|  *91.75* ($\downarrow$ 1.17)| 88.71 | 87.90 | *89.62* |
> |GNM-PT      |  **91.94** ($\downarrow$ 1.73)| **90.17**  | **88.21**  | **90.28**  |
>
> **Note:** The listed decreases in Head acc. are compared to that in Table R5.
>
>
> All methods, except for LPT, utilize the same backbones and training strategies, differing only in their optimization techniques. We directly cite the result of LPT from its original paper. Tables R5 and R6 demonstrate that GNM can further enhance model performance compared to CCSAM and imbSAM. Additionally, GNM requires less computation time than other SAM-based methods.
>
> ---
> > Q3. Can loss reweighting or logit adjustment alleviate the issue of flat minimum dominated by head classes?
>
> **A3:**  Literature indicates that logit adjustment alone is insufficient to fully address the challenge of a flat minimum predominantly influenced by head classes. LDAM and GCL are two representative methods of logit adjustment. As presented in CC-SAM [R1], the loss landscapes for LDAM (Figure 5 in [R1]) and GCL (Figure 8 [R1]) do not exhibit flattening compared to CE. Specifically, the loss landscape of LDAM is sharper than that of CE for all classes, while GCL, despite generally having a flatter loss landscape than CE, exhibits several small local protrusions.
>
>
> [R1]  Z. Zhou, et al., Class-conditional sharpness-aware minimization for deep long-tailed recognition, in *CVPR*, 2023.
>
> ---
> > Q4. Analysis of how the hyper-parameter influences the model performance.
>
> **A4:**  Thanks for pointing out this issue. We will provide a comprehensive analysis of the impact of the hyper-parameter on the performance in the revised version. The following is our analysis.
>
> * When $a \rightarrow 0$, the interference becomes negligible, effectively restricting the loss function to attain its minimum value within a small area. The extreme case is $a=0$, meaning no additional optimization techniques are employed. Therefore, the smaller $a$ is, the less pronounced its effect.
>
> * When $a$ increases: the disturbance area expands. A large $a$ introduces significant perturbations, potentially deviating from the basic gradient descent path. Excessively large values of $a$, for example $a=2$, lead to performance degradation. In the extreme case of $a=\infty$, the model fails to converge.

---

> > ### Comment · Reviewer_XQK2 · 2024-08-08
> >
> > Thanks for the response. Part of my problems have been solved. However, I find that GNM-PT cannot outperform imbSAM in the first stage but outperform in the second stage. The author may give some analysis of this phenomenon. Furthermore, why the GNM-PT can outperform previous SAM-based methods also needs to be analyzed to help us understand your method.
> >
> > In addition, the two-stage training procedure is tedious, and it is unsatisfactory that GNM-PT cannot surpass imbSAM in the first stage.

---

> ### Author Response · Authors · 2024-08-08
>
> Thank you for your time and continued discussion. Below, we provide our responses to your comments and hope they address your concerns.
>
> ---
> > Q1. Analysis for GNM-PT cannot outperform imbSAM in the first stage but outperform in the second stage.
>
> **A1:** In brief, stage 1 of imbSAM already applies strong regularization to tail classes, the rebalancing strategy in stage 2 essentially duplicates this effect. In contrast, GNM applies the same level of regularization to all classes without specifically intensifying it for tail classes and thus the strong regularization for tail classes can still work in stage 2.
>
> In detail, the rebalancing strategy can be viewed as a strong regularization for tail classes. ImbSAM filters the gradient restrictions on head and median classes and only exerts perturbation on tail classes. As shown in Table R5, the performance gain of imbSAM in stage 1 primarily comes from tail classes, while the head classes see a slight improvement, and the median classes even exhibit a decline in performance. In stage 2, the rebalancing strategy that focuses on training tail classes duplicates the effect of stage 1, resulting in relatively smaller performance improvements.
>
> Differently, GNM imposes constraints on minimizing the optimal point and its neighborhood of the loss function to all classes equally. It comprehensively improves the performance across all classes. Compared to imbSAM, GNM applies balanced regularization across all classes. Therefore, adding rebalancing can enhance overall performance by significantly improving the performance of the tail class.
>
> ---
> > Q2. Why the GNM-PT can outperform previous SAM-based methods.
>
> **A2:** In brief, for long-tailed learning, SAM tends to favor head classes during optimization, CCSAM and imbSAM are biased towards optimizing tail classes, while GNM offers a balanced optimization across all classes.
>
> For GNM: The optimization constraint in GNM is sample-independent, preventing classes with large sample sizes from dominating the direction of the perturbation vector. GNM can effectively improve all classes, avoiding optimization biases towards head classes and ensuring that the optimization focus is not solely on tail classes. Table R5 supports this as well.
>
> For SAM: The head class dominates gradient optimization as the perturbations are dependent on class size. Remark 1 (lines 176-182) and Appendix B (lines 528-542) provide a detailed analysis and proof.
>
> For CCSAM: It scales the perturbations in SAM utilizing a weight inversely proportional to the class sizes.
> For imbSAM: It directly filters out the perturbations from the head class and median classes in SAM, focusing solely on adding smooth constraints to the sharpness of tail classes.
> Both of these two methods prioritize the sharpness constraint of tail classes while overlooking the other classes.
>
> In addition, GNM has another significant advantage: it requires negligible computational overhead due to its single forward and backward pass per gradient update. In contrast, other SAM-based methods require an additional forward and backward pass per gradient update, and thus double the computation time. Remark 2 (lines 183-190 in the paper) provides a detailed analysis.

---

> ### Author Response · Authors · 2024-08-08
>
> **(Continued from previous)**
>
> ---
> > Q3. Two-stage training procedure is tedious.
>
> ** A3:** The two-stage method is one of the most widely used training strategies for long-tail learning [R1-R7]. It requires only a few additional epochs for the classifier to be trained and does not introduce significant overhead. DRW [R1] can even be trained end-to-end without requiring additional computational parameters, time, or other resources.
>
> ---
> > Q4. GNM-PT cannot surpass imbSAM in the first stage.
>
> **A4: ** We admit the necessity of a rebalancing strategy for GNM-PT is the limitation and have discussed it in the concluding remarks.
>
> GNM-PT does not apply strong regularization to tail classes, resulting in less improvement for these classes compared to imbSAM for one-stage training. Therefore, we apply an additional re-balance stage for strong regularization to tail classes and for further balancing the classifier.  GNM-PT demonstrates improvements across all class scales in Stage 1 and surpasses imbSAM after Stage 2.
>
> Computational overhead is also an important metric to consider. Although GNM-PT is slightly inferior to imbSAM in stage 1, it is worth noting that imbSAM requires 88.97 seconds per epoch due to the need for two forward and backward propagations. In contrast, GNM only requires 42.77 seconds per epoch, which significantly reduces the computational burden, as analyzed in Remark 2 (lines 183-190).
>
> Reference:
>
> [R1] K. Cao, et al., Learning imbalanced datasets with label-distribution-aware margin loss, in NeurIPS 2019.
>
> [R2] B. Kang, et al., Decoupling representation and classifier for long-tailed recognition, in ICLR 2020.
>
> [R3] Z. Zhong, et al., Improving calibration for long-tailed recognition, in CVPR 2021.
>
> [R4] M. Li, et al., Long-tailed visual recognition via gaussian clouded logit adjustment, in CVPR 2022.
>
> [R5] B. Dong, et al., LPT: long-tailed prompt tuning for image classification, in ICLR 2023.
>
> [R6] J.X. Shi, et al., How Re-sampling Helps for Long-Tail Learning? in NeurIPS 2023.
>
> [R7] M. Li, et al., Feature Fusion from Head to Tail for Long-Tailed Visual Recognition, in AAAI 2024.

---

### Official Review · Reviewer_ZADq · 2024-07-12

**Soundness:** 3
**Presentation:** 3
**Contribution:** 3
**Rating:** 5
**Confidence:** 4

**Summary:**

This paper proposes Gaussian neighborhood minimization (GNM) to enhance prompt tuning methods in long-tailed recognition. GNM is inspired by a recent work SAM, which aims to achieve flat minima by capturing the sharpness of loss landscape. Theoretical evidence shows that GNM can achieve a tighter upper bound and optimize the loss to a lower value. Experiments on multiple long-tailed datasets demonstrate that GNM can improve the performance of both head and tail classes and can reduce the computational cost compared to SAM.

**Strengths:**

1. The studied problem is important, i.e., long-tailed recognition by fine-tuning the pre-trained foundation model.
2. The proposed method GNM is simple while versatile.
3. The theoretical contribution of this work is important.
4. The empirical results are thorough and convincing.

**Weaknesses:**

1. In Tables 1-3, the SAM-based methods are equipped with DNN-based model, which is unfair when compared to GNM-PT. Have you experimented the SAM-based methods with MHSA-based models such as ViT-B/16?
2. Since you have already used the GCL loss function, why do you use DRW? Will this make the learning process overly focused on tail classes?
3. The performance gain is marginal in some datasets, for example, when compared to LPT on iNaturalist and Places-LT. There may be space for further refinement.

**Questions:**

See weaknesses above.

**Limitations:**

The authors have discussed the limitations in their paper.

---

> ### Author Rebuttal · Authors · 2024-08-06
>
> >Q1. Comparison with MHSA-based model incorporated with SAM-based methods.
>
> **A1:** We implement the experiments using CIFAR-100-LT with an imbalance ratio of 100. Since the re-balancing strategy employed in the second stage can influence the performance of optimization methods on a per-class level, we conducted a comparative analysis for various optimization methods, both without and with the application of the rebalancing strategy. DRW is utilized as the re-balancing strategy. LPT also employs two stages that include a re-balancing strategy, thus we present it in Table R4.
>
> **Table R3.** Optimization strategy comparison (w.o. stage2).
> |   Method                 |  Head Acc. (%)| Med. Acc. (%)| Tail Acc. (%)| All Acc. (%) |NET (s)
> | --------                 |-------| ----- |------ | -----|-----|
> |VPT (ViT-B/16) w. GCL     |  92.86 | *88.94* |79.28  | 87.76 |**40.32**|
> |VPT (ViT-B/16) w. GCL & CCSAM |  92.81 | 88.31  |79.31  | 87.84 |80.47|
> |VPT (ViT-B/16) w. GCL & imbSAM|  *92.92* | 88.43  |**84.00**  | **89.02** | 88.97|
> |VPT (ViT-B/16) w. GCL & GNM   |  **93.67** | **89.03**  |*81.10* | *88.46* | *42.77*|
>
> **Note:** NET represents native execution time.
>
> **Table R4.**  Optimization strategy comparison (w. stage2)
> |   Method       | Head Acc. (%)| Med. Acc. (%) | Tail Acc. (%) | All Acc. (%) |
> | --------       |-------| ----- |------ | -----|
> | LPT            | *     |   *   |     * |89.10 |
> |VPT (ViT-B/16) w. GCL  |90.08 ($\downarrow$ 2.78)|89.60|*88.14* | 89.40 |
> |VPT (ViT-B/16) w. GCL & CCSAM | 90.47 ($\downarrow$ 2.34)| *89.63* | 88.03 | 89.54 |
> |VPT (ViT-B/16) w. GCL & imbSAM|  *91.75* ($\downarrow$ 1.17)| 88.71 | 87.90 | *89.62* |
> |GNM-PT      |  **91.94** ($\downarrow$ 1.73)| **90.17**  | **88.21**  | **90.28**  |
>
> **Note:** The listed decreases in Head acc. are compared to that in Table R3.
>
> All methods, except for LPT, utilize the same backbones and training strategies, differing only in their optimization techniques. We directly cite the result of LPT from its original paper. Tables R3 and R4 demonstrate that GNM can further enhance model performance compared to CCSAM and imbSAM. Additionally, GNM significantly reduces computational overhead compared to other SAM-based methods.
>
>
> Table 8 also compares AdapterFormer with two different pre-trained ViTs under different optimizations. Please refer to the appendix for details.
>
> ---
> > Q2. Using GCL with DRW.
>
> **A2:** Thanks for pointing out this problem. We agree that incorporating a re-balancing strategy poses the risk of over-emphasizing tail classes. While employing only GCL mitigates the gradient over-suppression problem, it does not fully address the classifier bias stemming from the significant disparity in sample sizes between head and tail classes. As noted in the original GCL paper, a two-stage strategy that includes classifier re-balancing is implemented to further enhance overall performance. Similarly, LPT [R1] utilizes GCL-based loss. During stage 2 of LPT, group prompt tuning also incorporates a re-balancing strategy that combines class-balanced and instance sampling data. To facilitate end-to-end training and further balance the classifier, we adopt DRW.
>
> [R1] Bowen Dong, et al. LPT: long-tailed prompt tuning for image classification. In ICLR, 2023.
>
> ---
>
> > Q3. The performance gain is marginal in some datasets.
>
> **A3:** Thanks for this valuable comment. While improvements on some datasets may appear marginal, GNM-PT exhibits greater computational efficiency. For example, on iNaturalist-2018, GNM-PT requires 70 epochs without DRW and 80 epochs with DRW, compared to LPT’s 160 epochs—80 for shared prompt tuning and 80 for group prompt tuning.
>
> Moreover, achieving significant improvements becomes increasingly challenging as the performance on long-tailed data approaches that of balanced datasets. Take cifar100-LT as an example: we conducted an experiment utilizing VPT on balanced cifar-100. The classification accuracy on the validation set is 92.9\%, which can be regarded as the upper bound of cifar100-LT. GNM-PT achieves 90.3\% on cifar100-LT with an imbalance ratio of 100, a value remarkably close to this upper bound.
>
> According to your kind suggestion, we leave this as one of our future research priorities and will further explore ways to improve the proposed method.

---

> > ### Comment · Area_Chair_kFt5 · 2024-08-13
> > **Reminder -- please reply to rebuttal**
> >
> > Dear Reviewer ZADq,
> >
> > Thank you again for reviewing this paper. Since the reviewer-author discussion phase is closing soon, could you please read through all the reviews and see if the authors' responses have addressed your concerns?
> >
> > Best,
> >
> > AC

---

### Official Review · Reviewer_X9cb · 2024-07-16

**Soundness:** 3
**Presentation:** 4
**Contribution:** 3
**Rating:** 5
**Confidence:** 3

**Summary:**

The paper proposes an optimization approach called Gaussian neighborhood minimization prompt tuning (GNM-PT) for long-tailed visual recognition. Compared to sharpness-aware minimization (SAM), it excels in lower computational overhead, tighter upper bound for loss function and superior performance. GNM-PT utilize Gaussian neighborhood instead of the gradient direction of current parameters employed in SAM.

**Strengths:**

1, The idea of using Gaussian neighborhood to search for flat minima is novel.
2, PEFT methods optimized by GNM-PT achieve state-of-the-art performance in long-tailed benchmarks.

**Weaknesses:**

1, For the comparative method LDAM [1] mentioned in the paper, the authors did not provide experimental results.
2, The experimental results are insufficient. Specifically, this paper investigates the optimization strategies for long-tailed recognition under parameter-efficient fine-tuning. Accordingly, existing optimization strategies should be compared under the same training paradigm (i.e., PEFT) and the same backbone. For example, in Tables 1 to Table 3, the performance of “LPT+SAM”, “LPT+CCSAM”, and “LPT+IMBSAM” should be provided to illustrate the outstanding superiority of the proposed GNM-PT.
3, I wonder whether the variance of the Gaussian distribution has an impact on the results. The authors should provide ablation studies on the different choices of variance.
4, As illustrated in Section 3.2, GNM-PT improves the generalization performance of each category equally. However, the results presented in Table 2 and Table 3 show that GNM-PT achieves inferior performance than the baseline LPT for tail category, please explain the reason.
5, There are some typos: In the explanation of Eq. (2), what do “k” and “n” represent?

[1] Cao K, Wei C, Gaidon A, et al. Learning imbalanced datasets with label-distribution-aware margin loss[J]. Advances in neural information processing systems, 2019, 32.

**Questions:**

See weakness.

**Limitations:**

GNM-PT method must collaborate with the rebalancing strategy to ensure overall performance across all classes.

---

> ### Author Rebuttal · Authors · 2024-08-06
>
> > Q1. Experimental results with LDAM.
>
> **A1:** Thank you for pointing out this issue. GCL is a logit adjustment method with a rationale similar to LDAM. Since LDAM is one of the baseline methods for GCL, and GCL performs better than LDAM on long-tailed visual recognition, we chose to compare our method with more advanced methods, omitting the results of LDAM in the experiment. We will include these results in our revised version.
>
> ---
> > Q2. Comparison results w.r.t. optimization strategies under the same backbone.
>
> **A2:** We implement the experiments using CIFAR-100-LT with an imbalance ratio of 100. The re-balancing strategy employed in the second stage can influence the performance of optimization methods on a per-class level, we conducted a comparative analysis for various optimization methods, both without and with the application of the rebalancing strategy. DRW is utilized as the re-balancing strategy. LPT also employs two stages that include a re-balance strategy, thus we present it in Table R2.
>
> **Table R1.** Optimization strategy comparison (w.o. stage2).
> |   Method                 |  Head Acc. (%)| Med. Acc. (%)| Tail Acc. (%)| All Acc. (%) |NET (s)
> | --------                 |-------| ----- |------ | -----|-----|
> |VPT (ViT-B/16) w. GCL     |  92.86 | *88.94* |79.28  | 87.76 |**40.32**|
> |VPT (ViT-B/16) w. GCL & CCSAM |  92.81 | 88.31  |79.31  | 87.84 |80.47|
> |VPT (ViT-B/16) w. GCL & imbSAM|  *92.92* | 88.43  |**84.00**  | **89.02** | 88.97|
> |VPT (ViT-B/16) w. GCL & GNM   |  **93.67** | **89.03**  |*81.10* | *88.46* | *42.77*|
> **Note:** NET represents native execution time.
>
>
> **Table R2.**  Optimization strategy comparison (w. stage2)
> |   Method       | Head Acc. (%)| Med. Acc. (%) | Tail Acc. (%) | All Acc. (%) |
> | --------       |-------| ----- |------ | -----|
> | LPT            | *     |   *   |     * |89.10 |
> |VPT (ViT-B/16) w. GCL  |90.08 ($\downarrow$ 2.78)|89.60|*88.14* | 89.40 |
> |VPT (ViT-B/16) w. GCL & CCSAM | 90.47 ($\downarrow$ 2.34)| *89.63* | 88.03 | 89.54 |
> |VPT (ViT-B/16) w. GCL & imbSAM|  *91.75* ($\downarrow$ 1.17)| 88.71 | 87.90 | *89.62* |
> |GNM-PT      |  **91.94** ($\downarrow$ 1.73)| **90.17**  | **88.21**  | **90.28**  |
>
> **Note:** The listed decreases in Head acc. are compared to that in Table R1.
>
> As observed from Table R1, without the re-balancing strategy to adjust the classifier bias, imbSAM achieves better overall accuracy. However, imbSAM has little impact on the head and middle classes. Additionally, both ccSAM and imbSAM require two back propagations, thereby doubling the computation time. Compared to VPT with GCL, which does not include additional optimization, GNM incurs only a small computational overhead.
>
> Table 2 shows that the re-balancing strategy sacrifices a small amount of head class performance in exchange for significantly improving tail class performance. imbSAM essentially does not employ additional optimization for head classes, whereas CCSAM and GNM use additional optimization for all classes. However, compared to imbSAM, CCSAM and GNM result in a greater reduction in the accuracy of head classes. The optimization strategies may have a limited impact on the rebalancing training process. We will investigate this in detail in future work.
>
> Additional comparative results across more datasets will be included in the revised version.
>
> ---
> > Q3. Ablation studies on different choices of variance.
>
> **A3:** We appreciate the reviewer's valuable suggestion. Table R3 presents ablation studies on the different choices of variance. Additionally, we include the results of $\tilde{\varepsilon}$ using a uniform distribution within the range of [-1,1].
>
> **Table R3.** Ablation study w.r.t. variance on CIFAR-100-LTwithimbalanceratio=100.
> |   $\rho$ | 3   |  2   | 1   | 0.8  | 0.6  |0.4   | (1/3) |0.2 |(uniform distribution)|
> | -------- |---- |------| ----- |------| -----|-----|-----|-----|-----|
> |Acc. (\%) |90.03|89.99 | 90.14 |90.23 | 90.01 | 90.06 |(90.28)|90.12|90.17|
>
>
> Initially, we considered only a Gaussian distribution with a mean of 0 and a variance of 1/3, as it has a 99.7% probability of being within the range [-1, 1]. This allows us to conveniently control the size of $\tilde{\varepsilon}$ by adjusting the amplitude hyperparameter.
>
> Table R3 indicates that the variance impacts model performance. This finding demonstrates that, in addition to the amplitude of $\tilde{\varepsilon}$, the distribution also influences model performance and is worthy of further study.
>
> ---
> > Q4. The reason of GNM-PT achieves inferior performance than LPT for tail classes.
>
> **A4:** Stage 2 in LPT also adopts a re-balancing strategy that has a more substantial effect than optimization. This strategy results in a minor reduction in the performance of head classes while significantly enhancing the performance of tail classes. For iNat, LPT did not present the performance in head classes. The results we re-implement are: 59.54\% (Head Acc.), 76.65\% (Med. Acc.), 79.12\% (Tail Acc.), 75.98\% (All Acc.). The enhancement by stage 2 in LPT of tail class performance comes with a trade-off that may negatively impact the performance on head classes.
>
> Table 2 presents the results of GNM-PT with and without the rebalancing strategy. After implementing the rebalancing strategy, GNM-PT's performance in tail classes matches that of LPT (GNM-PT: 79.3\% v.s. LPT: 79.3\% on iNat, and GNM-PT: 49.4\% v.s. LPT: 48.4\% on Places-LT).
>
> In addition, comparing Table R1 with Table R2 reveals that the application of additional optimization on the rebalancing strategy in stage 2 may not consistently contribute to the improvement of model performance. This observation warrants further investigation, which we will study in our future work. We appreciate the reviewers' valuable insights that inspired this direction.
>
> ---
> > Q5. Typos.
>
> **A5:** Thank you for the reminder. We will proofread and revise the paper carefully.

---

> > ### Comment · Area_Chair_kFt5 · 2024-08-13
> > **Reminder -- please reply to rebuttal**
> >
> > Dear Reviewer X9cb,
> >
> > Thank you again for reviewing this paper. Since the reviewer-author discussion phase is closing soon, could you please read through all the reviews and see if the authors' responses have addressed your concerns?
> >
> > Best,
> >
> > AC

---

> > ### Comment · Reviewer_X9cb · 2024-08-13
> > **Response to Authors**
> >
> > Thanks for the rebuttal, and most of my concerns are addressed. I will keep my positive rating to this paper.

---

> ### Author Response · Authors · 2024-08-13
>
> Dear Reviewer X9cb,
>
> We are glad to hear that most of your concerns have been addressed. Thank you for your thoughtful consideration. Your positive rating is greatly appreciated. We hope that our paper can meet your expectations.
>
> Should you have any further concerns, please let us know and we are more than willing to engage in a detailed discussion before the discussion phase deadline. We are committed to addressing all feedback to the best of our ability to ensure the paper aligns with the high standards of the conference.
>
> Best regards,
>
> Authors

---

### Official Review · Reviewer_veiW · 2024-07-17

**Soundness:** 3
**Presentation:** 3
**Contribution:** 3
**Rating:** 6
**Confidence:** 4

**Summary:**

This work addresses the long-tailed learning problem by adding a tight upper bound on the loss function of data distribution and improving the generality of the model through flattening the loss landscape.

**Strengths:**

1. Long-tailed visual recognition is an inevitable problem and is desired for conducting in-depth research. The authors provide a new insight into tackling this issue.
2. This paper is well-written and easy-to-understand, and it includes theoretical justifications, detailed algorithm descriptions, and experimental results to support the claims.

**Weaknesses:**

1. The authors mention that GNM flattened the loss landscape and thus improved model generalization. I doubt this conclusion, since the flatness improvement from the proposed approach is not significant enough. Specifically, the original landscape is already flat enough in Figures 1 and 4 (i.e., compared with the original bumpy landscape in Figure 1 of [ref1] and Figure 1 of [ref2]). The original approach already possessed the ability to achieve generality without needing any more skills, and thus the GNM seems to contribute little. Hence, it comes to the confusion: does GNM really help the model improve generalization through this perspective (flattening the loss)?

2. I hope to have more quantitative metrics about the loss landscape. Since the only value in Figures 1 and 4 is loss, which only means lower loss buttum with higher accuracy (i.e., it is hard to compare the convexity of various approaches since they are both very flat), Could the author measure the level of convexity? For example, you can compute the eigenvalues of the Hessian following [ref2].

3. I noticed that the authors only chose to visualize the loss landscape of CIFAR100-LT. Could you clarify why you only chose this one? And does it have a similar phenomenon in the other two datasets? Could you provide more visualization results in the appendix?

[ref1] Sharpness-aware minimization for efficiently improving generalization. ICLR 2021
[ref2] Visualizing the Loss Landscape of Neural Nets. NeurIPS 2018

**Questions:**

See "Weaknesses”

**Limitations:**

The authors included the limitation section.

---

> ### Author Rebuttal · Authors · 2024-08-07
>
> > Q1. Does GNM help improve generalization through flattening the loss?
>
> **A1:** Thanks for pointing out this problem. Flattening the loss landscape is one aspect to consider. As observed in Figures 1 and 4, GNM exhibits a relatively large "area" at the minimum, particularly compared to the original method, although this may not be particularly obvious. Moreover, as discussed in Remark 4 (lines 219-225), GNM can achieve a tighter upper bound on the loss. Therefore, GNM further enhance model performance. We will revise the relevant content accordingly.
>
> ---
> > Q2. More quantitative metrics about the loss landscape.
>
> **A2:** Thank you for your constructive feedback. We compute the eigenvalues of the Hessian using CIFAR-100-LT. Calculating the eigenvalues of the Hessian matrix is time-consuming, requiring approximately 4300 seconds per point on an RTX 4090 GPU. Due to time constraints, we currently present 1D visualization results, as shown in Figure R1 in the attachment in "global rebuttal". (We also include a copy of the figures in the anonymous link provided in the abstract of our paper, in case the attached file cannot be viewed.)
>
> As shown in the figure, GNM exhibits generally smaller eigenvalues. Both methods, however, display instances of larger eigenvalues. This could be attributed to the selected one-dimensional direction and the limited number of sampling points. We will include additional higher-resolution results in the revised version.
>
> ---
> > Q3. Reasons of visualizing the loss landscape of CIFAR-100-LT and more visualization results.
>
> **A3:**
>
> *Utilizing CIFAR-100-LT for visualization:* Existing methods, such as [R1] and [R2], utilize the CIFAR dataset for visualization. We primarily follow their settings.
>
> [R1] Z. Zhou, et al., Class-conditional sharpness-aware minimization for deep long-tailed recognition, in *CVPR*, 2023.
>
> [R2] H. Li, et al., Visualizing the Loss Landscape of Neural Nets, in *NeurIPS*, 2018.
>
> *More results:*
> Owing to the time constraint, we present the visualization results for the places-LT trained with Resnet152, which are shown in Figure R2 in the attachment in "global rebuttal". As shown in Table 11 in the appendix of our paper (Section J), the performance differences are small, making the distinctions in Figure R2 not particularly obvious. However, it can still be observed that SGD and SAM exhibit steeper gradients and some protrusions.
>
> We will supplement more comparative visualization results with the other two datasets and more methodologies in the revised revisions.

---

> > ### Comment · Reviewer_veiW · 2024-08-07
> >
> > Thank you for the detailed responses. My concerns have been successfully addressed.
> >
> > In additional to my existing comments, I think it would be beneficial for the authors to include a discussion of [1][2] in the revision. [1] approaches VPT by considering the Transformer architecture, while [2] provides a thorough analysis of VPT and its usability.
> >
> > [1] E^ 2VPT: An Effective and Efficient Approach for Visual Prompt Tuning;
> > [2] Facing the Elephant in the Room: Visual Prompt Tuning or Full Finetuning?

---

> ### Author Response · Authors · 2024-08-08
>
> Dear Reviewer veiW,
>
> We would like to thank you again for your precious time and the insightful recommendations. In addition to the revisions already made, we will incorporate a detailed analysis and discussion of these two important works relevant to VPT.
>
> Moreover, please let us know if you have any further concerns. We are more than willing to discuss them with you and will address all concerns to the best of our ability to ensure the paper meets the high standards of the conference.
>
> Best regards,
>
> Authors

---

> > ### Comment · Reviewer_veiW · 2024-08-08
> >
> > I am glad to learn our discussion is helpful. I have made my final rating. Good luck!

---

> ### Author Response · Authors · 2024-08-08
>
> Dear Reviewer veiW,
>
> We sincerely appreciate your thorough review and recognition of our work. Your valuable suggestions will significantly contribute to improving the quality of our paper.
>
> Best regards,
>
> The Authors

---

### Author Rebuttal · Authors · 2024-08-07

We would like to express our sincere gratitude to the PCs, SACs, ACs, and all the reviewers for their effort to enhanc our work and for their positive feedback. For example, Reviewer veiW pointed that our work *"includes theoretical justifications, detailed algorithm descriptions, and experimental results to support the claims."* Reviewer ZADq commented that *"the idea of using Gaussian neighborhood to search for flat minima is novel"* and *"the proposed method GNM is simple while versatile"* and *"the theoretical contribution of this work is important."* Reviewer XQK2 also recognized that *"the proposed method is simple and effective"* and praised the *"solid theoretical supports"*. We are greatly encouraged by the acknowledgment of the significance of our work by all the reviewers.

The reviewers' comments are of great value for enhancing our paper and inspiring our future work. In the revised version, we will include more quantitative metrics about the loss landscape, provide visualization results of additional datasets, and supplement more comparison results in the appendix.

The attachment provides the visualization results for Q2 and Q3 raised by Reviewer veiW.

---

### Decision · Program_Chairs · 2024-09-25

**Decision:**

Accept (poster)

**Comment:**

This paper proposes an optimization strategy, namely Gaussian neighborhood minimization prompt tuning, to tackle the long-tail classification problem. The proposed method employs the Gaussian neighborhood loss to provide a tighter upper bound on the loss and facilitate a flattened loss landscape. It seeks the gradient descent direction within a random parameter neighborhood, and thus reduces the computational overhead compared to sharpness-aware minimization.

Initial concerns raised by reviewers include that the effectiveness of some key designs is not well supported, the evaluation of the loss landscape is insufficient, there lacks several aspects in the experimental comparisons, and the influence of hyperparameters is not analyzed, The authors’ rebuttal well addressed most of these concerns and all reviewers were positive in their final recommendations. The AC agrees with the reviewers that the proposed method is simple yet effective, supported by theoretical analysis. Therefore, the AC recommends to accept. Reviewers did raise valuable concerns that should be addressed. The camera-ready version should include the discussions and results provided in the rebuttal.